# Trustless Audits without Revealing Data or Models

## Abstract

There is an increasing conflict between business incentives to hide models and data as trade secrets, and the societal need for algorithmic transparency. For example, a rightsholder wishing to know whether their copyrighted works have been used during training must convince the model provider to allow a third party to audit the model and data. Finding a mutually agreeable third party is difficult, and the associated costs often make this approach impractical.

In this work, we show that it is possible to simultaneously allow model providers to keep their model weights (but not architecture) and data secret while allowing other parties to trustlessly audit model and data properties. We do this by designing a protocol called ZkAudit in which model providers publish cryptographic commitments of datasets and model weights, alongside a zero-knowledge proof (ZKP) certifying that published commitments are derived from training the model. Model providers can then respond to audit requests by privately computing any function $F$ of the dataset (or model) and releasing the output of $F$ alongside another ZKP certifying the correct execution of $F$. To enable ZkAudit, we develop new methods of computing ZKPs for SGD on modern neural nets for simple recommender systems and image classification models capable of high accuracies on ImageNet. Empirically, we show it is possible to provide trustless audits of DNNs, including copyright, censorship, and counterfactual audits with little to no loss in accuracy.

## 1 Introduction

As ML models become more capable, businesses are incentivized to keep the model weights and datasets proprietary. For example, Twitter recently released their algorithm but not the model weights Twitter (2023), and many LLM providers only provide access via APIs. On the other hand, there is also an increasing societal need for transparency in the model and data behind these APIs: closed models harm transparency and trust Bell (2023).

To address this, we want to perform audits where model providers and users agree on specific properties to test for the ML training procedure and models. For example, the property that the training dataset contains no copyrighted content or a recommender system is not censoring items (e.g., tweets). An audit would ideally release results for exactly these properties and nothing else.

Currently, there are three methods of performing such audits on modern ML methods. One method is to release the data, random seed, and final model weights: the user can replay training. However, this procedure does not keep the data and weights hidden. Another method is multi-party computation (MPC), which allows several parties to participate in a computation while keeping data hidden. Unfortunately, MPC requires all participants to participate honestly, but an audit presupposes a lack of trust between the model provider and users. MPC that handles malicious adversaries is extremely bandwidth intensive: a back-of-the-order calculation suggests the training procedure for an *8-layer CNN* may take up to *5 petabytes* of communication, which would cost $450,000 in cloud egress fees Pentyala et al. (2021). Finally, a trusted third party (TTP) could perform the audit, but TTPs are rarely practical. The TTP must have access to trade secrets, which model providers wish to keep secret. Furthermore, audits are expensive, requiring highly specialized expertise (deep understanding of ML training), and strong security (to avoid leaking trade secrets). In many cases, no viable TTPs are trusted by both model providers and users.

Prior work has proposed using zero-knowledge proofs to perform audits to address this issue Kroll (2015); Shamsabadi et al. (2022). Zero-knowledge proofs allow a prover to prove properties about their data (e.g., training data or model weights) without revealing the data itself. However, none of this prior research extends to modern ML methods in the form of deep neural networks (DNNs).

In this work, we develop an auditing procedure ZKAUDIT that can perform audits *without third parties and without any assumptions of trust* (i.e., trustlessly) on modern DNNs. ZKAUDIT, via zero-knowledge proofs, allows a model provider to selectively reveal properties of the training data and model without a TTP such that any party can verify the proof after the fact (i.e., the audit is non-interactive). Importantly, these guarantees are *unconditional*, providing security against malicious adversaries with only standard cryptographic assumptions.

ZKAUDIT consists of two steps: ZKAUDIT-T and ZKAUDIT-I. In ZKAUDIT-T, the model provider trains a model and publishes a commitment (e.g., hash) of their dataset, model weights, and a zero-knowledge proof that proves the weights were generated by training on the committed dataset. Then, in ZKAUDIT-I, the user provides an arbitrary audit function $F$. The model provider executes $F$ on the *same* weights and dataset used in training and provides a zero-knowledge proof of the execution. The zero-knowledge proof guarantees that $F(\text{data,weights})$ was executed on the hidden model/weights and evaluated honestly. For example, $F$ could check whether a training set contains copyrighted data or whether a social media provider is shadowbanning posts. The model provider can trustlessly evaluate $F$ by using prior work to generate zero-knowledge proofs for inference Lee et al. (2020); Weng et al. (2022); Feng et al. (2021); Kang et al. (2022).

To enable ZKAUDIT, we leverage recent development in cryptographic techniques known as ZK-SNARKS (zero-knowledge succinct non-interactive argument of knowledge). ZK-SNARKS allow a prover to produce a proof that an arbitrary computation happened correctly (Section 2). However, ZK-SNARKS are incredibly costly: it can take up to days to prove the execution of the forward pass on even toy models Lee et al. (2020); Weng et al. (2022); Feng et al. (2021). Only recently has it become possible to produce proofs of the forward pass on real-world models Kang et al. (2022). However, no existing work can compute the backward pass necessary for gradient descent, a requirement for proofs of training.

To produce proofs of training, we extend recent work to compute the backward pass of real-world DNNs. Our work enables model providers to produce proofs of full stochastic gradient descent on private data. Doing so requires overcoming several challenges: prior work uses integer division and `int8` precision for efficient forward pass computation. Unfortunately, training with these settings is not amenable to achieving high accuracy. We provide methods of embedding stochastic gradient descent with rounded division and variable fixed-point precision, and show that training in fixed-point can achieve high accuracy.

On commodity hardware, ZKAUDIT can produce audits of image classification systems and simple recommender systems with little to no loss in accuracy on a range of real-world datasets (medical datasets and standard benchmark datasets). The cost of auditing a recommender system and image classification system can be as low as $10 and $108, respectively, showing the practicality of our work. Achieving these low costs requires all of our optimizations: training would suffer dramatic losses in accuracy or not proceed without them.

## 2 BACKGROUND ON ZK-SNARKS

**ZK-SNARKs.** ZK-SNARKs are a cryptographic primitive that allows a *prover* to produce a proof $\pi$ that some function $F(x,w)$ was computed correctly, where $x$ is public and $w$ is private. Given $\pi$, a verifier can check that the prover computed $F$ correctly without access to $w$.

ZK-SNARKs have several amazing properties. First, they are *succinct*, i.e., small in the size of the input. Second, they are *non-interactive*, meaning the prover and verifier need not interact beyond $\pi$. Third, they are *knowledge sound*, which means that a computationally bounded prover cannot generate proofs for incorrect executions. Fourth, they are *complete*, meaning that proofs of correct execution verify (often unconditionally). Finally, they are *zero-knowledge*, which means $\pi$ reveals nothing about the private inputs beyond what the output and public inputs contain.

Although ZK-SNARKs allow arbitrary computation to be proved, ZK-SNARKs require computation to be expressed in specific ways. The cryptography community has provided several such ways of expressing computations, including R1CS Groth (2016) and Plonk Gabizon et al. (2019).

Unfortunately, naively expressing computations can result in highly inefficient proof generation. The specification of the computation and proving systems can jointly result in *three orders of magnitude* or more differences in proving times.

**Representing computation.** We describe salient details of representing computation in ZK-SNARKs. Although other works describe relevant details, it is critical to understand the basic building blocks and costs associated with computation in ZK-SNARKs to understand our optimizations.

In this work, we leverage arithmetic intermediate representations (AIRs), represented by a 2D grid $x_{ij}$ of values. We denote the number of rows as $R$ and columns as $C$. Due to the construction of ZK-SNARKs, the $x_{ij} \in \mathbb{F}_q$ for some large prime $q$. In particular, arithmetic is done *in the finite field*, so standard operations such as division are not natively possible.

Logically, there are three ways to constrain values on the grid:

1. Constraining two values to be equal: $x_{ij} = x_{i'j'}$.

2. Constraining a subset of a row to be in a pre-defined table: $(x_{ij_1},...,x_{ij_k}) \in \{(t_1,...,t_k)\} = T_m$ for some table $T_m$. $T_m$ is called a lookup table.

3. A polynomial constraint on a grid row: $f_l(x_{i1},...,x_{iR}) = 0$ for some polynomial $f_l$.

We provide an example of using polynomial constraints to implement integer division in Section 4, and Kang et al. (2022) provide other examples. In general, the costs increase with the number of rows ($R$), number of columns ($C$), maximum degree of the polynomial constraints $f_l$, and number of lookup tables ($T_m$). Furthermore, the number of rows must be a power of two. Given a representation of a computation in an AIR, we can produce a concrete ZK-SNARK proof by using a proving system such as halo2 zcash (2022).

We provide an extended discussion of ZK-SNARKs in Appendix A, including examples of using AIRs and how to compile ZK-SNARKs.

## 3 ZKAUDIT: PRIVATE AUDITS OF ML

**Protocol.** We describe ZKAUDIT when given access to verified randomness, a public source of timestamped, verified random bits. The need for verified random bits can be removed with the slightly stronger assumption of a random oracle hash function, which we describe in Appendix B.1. Throughout, we assume access to a binding and hiding commitment scheme, in which the trainer commits to the training data and cannot change the values later. The commitment scheme can be implemented in practice by publicly releasing hashes of the data and weights.

The first part of ZKAUDIT (ZKAUDIT-T) proves that the trainer honestly trained a model with known architecture but hidden weights on a hidden dataset. To do so, the trainer commits to the data, commits to a training order, and produces ZK-SNARKs for SGD from a randomly initialized or public pre-trained model:

1. The trainer commits to a dataset $\{(x_1,y_1),...,(x_n,y_n)\}$, producing commitments $[c_1,...,c_n]$. The commitments are ordered lexicographically, and the trainer publicly posts the commitments.

2. The trainer uses a verified source of randomness to generate a traversal ordering of the dataset (see Appendix B for why this is desired in some circumstances).

3. The trainer computes ZK-SNARKs of the SGD process, one batch at a time, using the traversal ordering. To do so, it computes the ZK-SNARK of the forward pass of any frozen layers, the forward and backward pass of any layers being updated, and the weight update procedure. This can be done in one or more ZK-SNARKs.

4. The trainer publishes the ZK-SNARKs of SGD and the commitment to the model weights at the end of training.

The second part of the protocol (ZKAUDIT-I) computes the zero-knowledge proof(s) for the audit function itself. Given the commitments to the dataset and final weights, the user sends an audit function $F(\text{data},\text{weights})$. The trainer then computes a ZK-SNARK of the audit function and publishes it (along with the output of $F$). For example, the audit may be that a recommender system is not censoring social media content. The model trainer must also hash the weights in the zero-knowledge proof to ensure trained weights from ZKAUDIT-T are consistent with the weights in ZKAUDIT-I.

**Security analysis.** ZKAUDIT has the following (informal) properties: 1) the trainer cannot "cheat" in training or computing the audit function, and 2) the verifier learns nothing about the training data and model weights aside from the output of the audit function. We can formalize these properties as knowledge soundness and zero-knowledge.

We provide a formal analysis of security (knowledge soundness and zero-knowledge) in Appendix B and provide an informal analysis here. For our security analysis, we assume two standard cryptographic primitives: a cryptographically secure hash function (informally, one that is secure against collisions) Menezes et al. (2018) and ZK-SNARKs Bitansky et al. (2017). It is standard to denote the security of these primitives with a parameter $\lambda$. Informally, the security parameter controls the probability that an adversary can "break" the protocol.

Denote the dataset size as $D$ and the number of stochastic gradient steps as $T$. Then, the prover produces at most $D + 4T$ hashes, commitments, and ZK-SNARKs. The security of each hash and ZK-SNARK follows directly from the primitives.

By the union bound, in order to achieve a security parameter of $\lambda$ for ZKAUDIT, we must choose the hash function and ZK-SNARK parameters so they have at least $(D+4T)\lambda$ bits of security.

**Security of ZK-SNARKs.** ZKAUDIT's security rests on the security of the underlying ZK-SNARK proving backend that is used (halo2 zcash (2022) in this work). Our ZK-SNARKs can be constructed via KZG commitments Kate et al. (2010) or inner-product arguments (IPA) Bünz et al. (2021). In the KZG version, we require a *structured-reference string* (SRS) that is universal to *all* audit functions. Namely, the SRS need only be generated once in a secure manner. To do so, we can use the already-generated SRS, which was generated using a perpetual trusted setup in which many parties participate (over 75 at the time of writing) PSE (2023). Only a single party needs to be honest for the setup to be secure. IPAs do not require any trusted setup.

**Limitations.** Although ZKAUDIT provides computational security against malicious adversaries and traversal ordering attacks, it has two major limitations. First, it does not protect against data poisoning attacks, in which a malicious attacker can manipulate the data itself Steinhardt et al. (2017). Second, while ZKAUDIT does not reveal the weights, it does reveal the model architecture. We view addressing these limitations as exciting future research.

## 4 COMPUTING ZK-SNARKS FOR GRADIENT DESCENT

We now describe our method and optimizations for computing gradient descent within a ZK-SNARK. Unlike in the computation of the forward pass, the input to gradient descent is both the input data and the model weights. The output is an updated set of weights. Formally, for an input $x$ and weights $w$, gradient descent computes $w' = G(x, w)$, where $w'$ is the updated set of weights. One standard method of performing gradient descent is to compute the forward pass, compute the backward pass, and update the weights by scaling the gradients by the learning rate.

Prior work has optimized the forward pass for `int8` inference in ZK-SNARKs Kang et al. (2022). In this work, we extend this prior work by showing how to compute the backward pass in a ZK-SNARK. We further optimize gradient descent by designing a high-performance softmax in ZK-SNARKs and operating in fixed-point arithmetic.

We first observe that the backward pass can often be expressed in structurally similar computation as the forward pass. For example, the backward pass of a convolution can also be expressed as a convolution (with different inputs).

However, several significant differences between inference and training necessitate changes.

**Rounded division and fixed-point.** Training requires more precise arithmetic than inference. For efficiency, prior work Kang et al. (2022) uses the floor function for `int8` arithmetic, which would result in poor accuracy for training. To understand why, consider the update formula for SGD: $w' = w + \eta \cdot \Delta w$ Typically, the learning rate $\eta$ is small (e.g., 0.01). When using lower precision, the multiplication by $\eta$ can be imprecise, leading to poor accuracy.

Thus, in order to compute ZK-SNARKs for gradient descent, we introduce two techniques: rounded division in finite field ZK-SNARK constraints and variable precision fixed-point arithmetic. Both techniques increase the accuracy of training.

We first implement rounded division in finite fields with polynomial constraints. As we show (Section 5), using rounded division can improve accuracy by up to 11% compared to standard integer division. We first describe how to implement standard integer division (which rounds towards zero). Suppose that $a,b,c,r$ are all positive. If $b = \lfloor \frac{a}{c} \rfloor$ then we have that the following constraint holds

$$a = b \cdot c + r \tag{1}$$

where $0 \le r < c$. To implement standard integer division, we first assume that $0 \le b,c < 2^N$ for some $N$. We can then use the polynomial constraint in Equation 1, constrain that $b,c,r \in \{0,...,2^N - 1\}$, and that $c - r \in \{0,...,2^N - 1\}$. Constraining that $c - r \in \{0,...,2^N - 1\}$ is equivalent to the constraint that $c > r$.

To implement rounded division, consider $a,b,c,r$ all positive integers. As before, we assume that $0 \le b,c < 2^N$ for some $N$. Let $b = \lceil \frac{a}{c} \rceil$. Then, the following constraints specify rounded division

$$2a + c = 2c \cdot b + r \tag{2}$$

where $0 \le r < 2c$. This follows because

$$b = \left\lfloor \frac{2a + c}{2c} \right\rfloor = \left\lfloor \frac{a}{c} + \frac{1}{2} \right\rfloor.$$

We can use similar constraints: Equation 2, a constraint that $b,c \in \{0,...,2^N - 1\}$, and a constraint that $2c - r \in \{0,...,2^{2N} - 1\}$. Although this requires a lookup table of size $2^{2N}$, we can implement rounded division, which is critical for training.

We further implement variable precision fixed-point arithmetic to allow trade-offs between accuracy and computation. Fixed-point arithmetic approximates real numbers by $\hat{x} = \text{Round}(x \cdot \text{SF})$, where SF is the scale factor. Since we use lookup tables to allow for non-linearities and fixed-point rescaling, more precision (i.e., a larger scale factor) results in larger lookup tables. This directly results in higher proving times but allows the model trainer to decide which precision level to use.

**Softmax.** In order to perform classification, we designed a high-performance softmax in ZK-SNARKs. To understand the difficulties of implementing softmax with finite field operations, recall the explicit formula: $y_i = \frac{e^{x_i}}{\sum_j e^{x_j}}$. Denote $s = \sum_j e^{x_j}$ and $\hat{e} = [e^{x_i}]$. Naively computing the softmax using the operations present would compute the exponential with a lookup table *in the standard units by scaling by the scale factor*, sum $e^{x_i}$, then divide by $s$. However, we must address three challenges when using fixed-point arithmetic to compute the softmax: underflow, precision, and range.

To understand these issues, consider a toy example where $x = [\ln \frac{1}{2}, 0]$, so $\hat{e} = [\frac{1}{2}, 1]$, $s = \frac{3}{2}$, and $y = [\frac{1}{3}, \frac{2}{3}]$ in full precision. Consider using a scale factor of 1000 for simplicity. The first issue arises in dividing by $s$: in the scaled units, $\hat{e} = [500, 1000]$, so $s = 1500$. However, *when dividing in the scaled units*, $y = [0,1]$. Thus, naive computation would result in a substantial loss in precision (underflow).

We can address the issue of underflow by dividing $s$ by the scale factor. However, this results in $s = 2$ and $y = [250, 500]$ in scaled units or $[\frac{1}{4}, \frac{1}{2}]$. This results in a relative error of *33%* in $y$, a substantial degradation in accuracy. In order to address this, we can scale $e^{x_i}$ by the scale factor again and not divide $s$ by the scale factor.

Finally, we use a standard trick to increase the numeric stability of the softmax. Since the softmax is shift-invariant, subtracting the maximum value results in a smaller range of outputs of the exponentiation. To compute the maximum of a vector, we can compute the pairwise maximum sequentially. In order to compute the pairwise maximum $c = \max(a, b)$ efficiently, we can use the following constraints. First, we constrain that $c$ is one of $a$ or $b$ by using the polynomial constraint $(c - a) \cdot (c - b) = 0$. We then constrain that $c - a, c - b \in [0,...,2^N)$, where $2^N$ is the size of our lookup table. This enforces that $c \ge a,b$.

| Scale factor | Proving time | Verification time |
|---|---|---|
| $2^{12}$ | 47.5 s | 10.0 ms |
| $2^{13}$ | 87.8 s | 9.9 ms |
| $2^{14}$ | 167.0 s | 9.9 ms |
| $2^{15}$ | 328.3 s | 9.8 ms |

Table 1: Proving and verification time of SGD on scale factors for image classification on a single image (MobileNet v2 (1.0, 224)). The proof size was 9.03 kb for all configurations.

| Scale factor | Proving time | Verification time |
|---|---|---|
| $2^{11}$ | 3.16 s | 6.2 ms |
| $2^{12}$ | 5.54 s | 6.1 ms |
| $2^{13}$ | 10.49 s | 6.3 ms |
| $2^{14}$ | 23.79 s | 6.0 ms |

Table 2: Proving time and verification time of SGD on a variety of scale factors for a recommender system (single example). The proof size was 4.6 kb for all configurations.

| Method | Proving lower bound |
|---|---|
| Zen | 200,000* s |
| vCNN | 172,800 s |
| pvCNN | 31,011* s |

Table 3: Estimated lower bounds for proving times of prior work for image classification on a *single* image. We exclude zkCNN since the authors explicit state that they are unable to compute the softmax Liu et al. (2021), so are unable to compute proofs of SGD.

## 5 EVALUATION OF ZKAUDIT-T

We now evaluate ZKAUDIT-T, including the performance of performing SGD in ZK-SNARKs, the end-to-end accuracy and costs of ZKAUDIT-T, and the effect of our optimizations. Because verification of ZK-SNARKs is cheap (10ms per proof), we focus on the cost of proving, which far dominates, and the accuracy (since there are potential degradations when discretizing).

We benchmarked SGD and ZKAUDIT-T on image classification and a recommender system on Movie Lens Harper & Konstan (2015). For the image classification tasks, we used a variety of MobileNet v2 configurations. The MobileNet configurations are denoted by the depth multiplier and input resolution, so MobileNet (1.0, 224) is a MobileNet v2 with a depth multiplier of 1.0 and an input resolution of 224×224. For the recommender system, we used a small model based on the DLRM model from Facebook Naumov et al. (2019). The complete configuration is in Appendix D.1.

To generate the cost estimates, we multiplied the total computation time by the cost of using a cloud computing platform (AWS) to perform the computation (Appendix D.1). We further conducted experiments on CIFAR-10 in the Appendix.

### 5.1 PERFORMANCE OF SGD

We first investigated the performance of embedding the computation of a single SGD step in a ZK-SNARK. We measure the proving time, verification time, and proof size.

We show results for image classification in Table 1 and for the recommender system in Table 2. The verification costs for ZK-SNARKs of SGD are incredibly low: as low as 6.0 ms. The proving times range from 26s to 328s for image classification and 2 s to 48s for the recommender system. Furthermore, none of the prior work implements the softmax operation, making SGD infeasible with this work. Nonetheless, comparing the proving times of our work to the proving times of just the arithmetic operations of prior work shows that our work is at least $95\times$ faster (Table 3).

### 5.2 END-TO-END ACCURACY AND COSTS

We then benchmarked the end-to-end accuracy of fine-tuning and costs. To do so, we chose three image classification datasets and one recommender system dataset. The image classification datasets ranged in task complexity, number of examples, and classes. We used the following datasets:

1. `dermnet` Shanthi et al. (2020): a dataset of skin images, where the task was to determine which one of 23 diseases the image belonged to. There were 15,557 training images.

2. `flowers-102` Nilsback & Zisserman (2008): a dataset of flower images, where the task was to classify images into one of 102 flower classes. There were 1,020 training images.

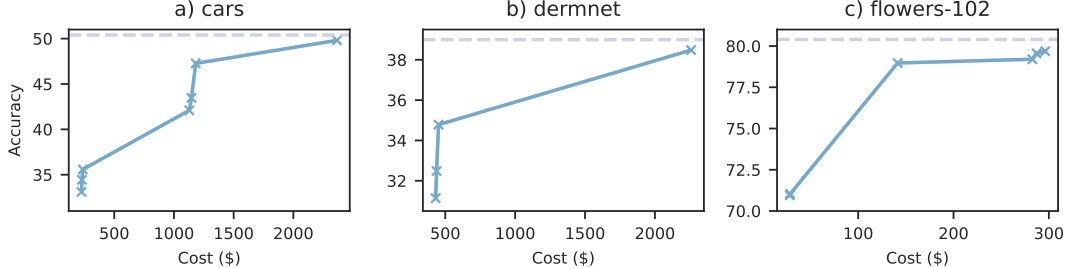

Figure 1: Test accuracy vs cost of proving training across the entire dataset for the Pareto frontier of image classification. Higher is better. The dashed line is the `fp32` accuracy.

| Dataset | Accuracy (fixed-point) | Accuracy (`fp32`) | Difference |
|---|---|---|---|
| `dermnet` | 38.5% | 39.0% | -0.5% |
| `flowers-102` | 79.7% | 80.4% | -0.7% |
| `cars` | 49.8% | 50.4% | -0.6% |

Table 4: Test accuracy of training with ZKAUDIT-T compared to full `fp32` accuracy. The loss in accuracy is marginal across datasets.

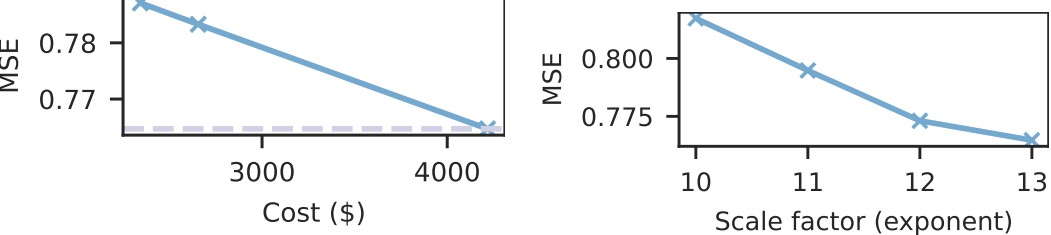

Figure 2: Test MSE vs total training cost for the Pareto frontier for the recommender system. Lower is better.

Figure 3: Test MSE vs scale factor. ZKAUDIT-T achieves parity with `fp32` at $2^{13}$.

3. `cars` Krause et al. (2013): a dataset of car images, where the task was to classify cars into one of 196 categories. There were 8,144 training images.

4. `movielens` Harper & Konstan (2015): a dataset of users ranking movies in IMDB. The training set has 6,040 users, 3,706 movies, and 900,188 ratings.

We estimated the costs of end-to-end verified training using the ZKAUDIT-T protocol by performing the full training procedure and estimating the cost of constructing ZK-SNARKs of the training run. We used a variety of MobileNet configurations and hyperparameters for image classification. We fixed the architecture for the recommender system but varied the hyperparameters.

We show the Pareto optimal frontier of accuracy and cost for the three image datasets in Figure 1 and the mean-squared error (MSE) for the recommender system in Figure 2. As shown, ZKAUDIT-T can smoothly trade off between accuracy and costs. Furthermore, ZKAUDIT-T can achieve high accuracy on all four datasets despite using fixed-point arithmetic.

Although the costs are high, practitioners can trade off between accuracy and proving costs. For example, privacy is required in a regulated medical setting, so the model cannot be revealed. However, for regulatory reasons, the model provider may desire to provide a transcript of training. In this setting, the model provider may want to achieve as high accuracy as possible. However, for some settings where small amounts of accuracy can be traded off for costs, a model provider can use ZKAUDIT-T for as little as $282 within 1% of `fp32` accuracy. Furthermore, we substantially improve over prior work. Even ignoring the softmax, the cost of the next cheapest method would be *$26,637* or 94× higher.

We further compare the accuracy when using standard `fp32` training. As shown in Table 4, the accuracy is close to the full precision counterpart. The recommender system's mean-squared error is on parity with full `fp32` accuracy.

| Model | Accuracy (`int` division) | Accuracy (rounded, ZKAUDIT) | Difference |
|---|---|---|---|
| MobileNet, 0.35, 96 | 59.1% | **70.4%** | 11.3% |
| MobileNet, 0.5, 224 | 75.7% | **86.3%** | 10.6% |
| MobileNet, 0.75, 192 | 79.2% | **88.8%** | 9.6% |

Table 5: Test accuracy (top-5) of models used by Kang et al. (2022) on ImageNet with rounded vs integer division. Integer division considerably hurts accuracy, indicating worse downstream fine-tuning.

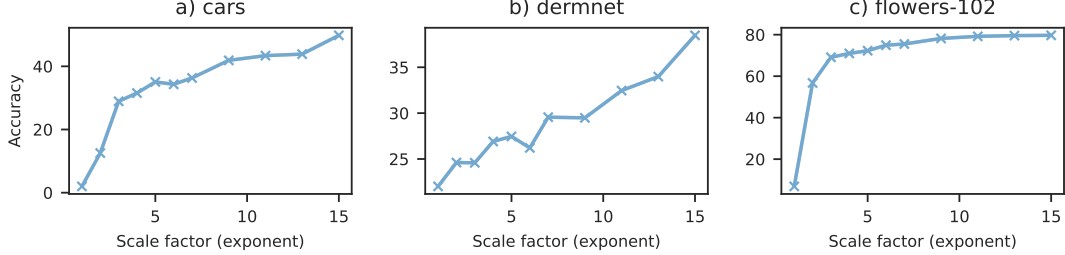

Figure 4: Test accuracy vs scale factor. As shown, we can achieve within 0.7% accuracy compared to full precision with a scale factor of $2^{15}$. The accuracy degrades with lower scale factors.

## 5.3 EFFECTS OF OPTIMIZATIONS

We investigated the effects of our optimizations: our improved softmax, rounded division, and precision. To do so, we removed our optimized softmax, removed rounded division, and reduced the precision (separately) to see the effects.

Removing our optimizations for softmax resulted in failure to train, as the range of the intermediate values was outside of the feasible range in the ZK-SNARK. Furthermore, no other work in ZK-SNARKs can perform the softmax, making training infeasible.

We then changed standard rounded division to integer division (rounded down) and computed the accuracy of the models on ImageNet as used by Kang et al. (2022). As shown in Table 5, the accuracy can drop as much as 11.3%. Since lower accuracy on ImageNet indicates lower performance for fine-tuning, we used rounded division for all further experiments.

We then reduced the training precision from a scale factor of $2^{15}$ to 1 for the image datasets. We reduced the training precision for the recommender system dataset from $2^{13}$ to $2^{10}$. We show results for the image datasets in Figure 4 and for the recommender system in Figure 3. As shown, the accuracy drops with lower precision. Our results corroborate results in low-precision training De Sa et al. (2017). Nonetheless, ZKAUDIT-T can achieve near-parity with `fp32` training.

These results show that our optimizations are necessary for high performance in ZKAUDIT-T.

## 6 USING ZKAUDIT FOR AUDITS

In addition to evaluating the ZKAUDIT training procedure, we describe and evaluate end-to-end audits and costs. In principle, ZKAUDIT is capable of performing any computable function, but we focus on audits of broader interest. For example, if the audit simply reveals the training data (which is a computable function), the model provider may choose to not participate. Our examples are meant as proof of concepts, and must be combined with work from the transparency literature for a full end-to-end solution. Our work focuses on the technical feasibility of privacy-preserving proofs of training and computability of audits. We describe how to perform an end-to-end audit in Appendix F.

Consider the case of recommender systems. Consumers of ML systems are interested in a wide range of audits. They may be interested in checking if certain items (e.g., Tweets or products) are censored Pesce (2023). A more extensive audit may also attempt to understand *counterfactual* behavior, in which the inputs to the recommender system are changed, and the effects on the recommendations are measured Akpinar et al. (2023). Outside of recommender systems, a copyright holder may wish to check that a model provider did not use their work in training or an auditor may perform a demographic disparity check. These audits require model outputs and a similarity check. Each audit

requires executing a different function, and as a result has a different cost profile, which we describe for each audit. We explore these audits below.

**Censorship audit.** To perform the censorship audit, we are interested if an item $x$ is ranked lower than the value implied by the recommender system. In particular, we are interested in whether an item the user believes should have a high ranking is censored. We can use random sampling to determine the quantile of the item $x$ among the full set of items or a subset of items previously shown to the user (ZKAUDIT-I). Determining the quantile is equivalent to estimating the parameter of a Bernoulli random variable and has rate $O(1/\sqrt{N})$. We use the Hoeffding bound to achieve finite sample estimates.

We executed this audit on the `movielens` dataset for estimating the quantile within 5% and 1% (600 and 14,979 samples, respectively). The true difference in quantile was 1.1% and 0.1%, respectively: the Hoeffding bound is known to be loose in practice Lee (2020). The costs were $0.42 and $10.59 for 5% and 1%, respectively, which are well within reason for many circumstances.

**Counterfactual audit.** The most comprehensive counterfactual audit measures the impact of interventions on recommender systems Akpinar et al. (2023). These interventions can be replacing inputs or changing hyperparameters. In order to do this audit, we can perform training twice and then estimate quantities. The total cost is twice the cost of training and the cost of estimating a quantity.

We performed the audit on the `movielens` dataset. We used a scale factor of $2^{13}$, which achieves parity with `fp32` accuracy (see above). The total cost was $8,456. To contextualize this result, the average cost of a financial audit of an S&P 500 is *$13,000,000* Analytics (2022). The full counterfactual audit would be only *0.07%* of the cost of a financial audit.

**Copyright audit, demographic disparity.** In the copyright audit, we prove with ZK-SNARKs that extracted features for each item (e.g., image) in the training set is dissimilar to features from a copyright holder's item. For the demographic disparity audit, we computed the demographic of each item and computed summary statistics. Both audits (from the perspective of ZK-SNARK computations) are the same cost. We performed these audit on the `flowers-102` dataset. The total cost of the audit was $108 (or about 10 cents per image), showing the feasibility of audits.

## 7 RELATED WORK

**Secure ML.** Recent work in the ML and cryptography literature has focused on the secure ML paradigm Ghodsi et al. (2017); Mohassel & Zhang (2017); Knott et al. (2021). Much of this work focuses on secure inference, in which a model consumer offloads computation to a service provider. The model consumer desires either privacy or validity. The techniques for secure ML range from multi-party computation (MPC) Knott et al. (2021); Kumar et al. (2020); Lam et al. (2022), zero-knowledge or interactive proofs Lee et al. (2020); Weng et al. (2022); Feng et al. (2021); Kang et al. (2022), and fully homomorphic encryption (FHE) Lou & Jiang (2021); Juvekar et al. (2018).

In this work, we focus on *training* as opposed to inference, with *malicious* adversaries. We provide the first fully-private training scheme for realistic datasets in the face of malicious adversaries.

**ZK-SNARKs.** Recent work has optimized ZK-SNARKs for DNNs Lee et al. (2020); Weng et al. (2022); Feng et al. (2021); Kang et al. (2022) and numerical optimization problems Angel et al. (2022). None of the prior work demonstrates how to perform the softmax and backward pass, both of which are required for training. In this work, we leverage ideas from inference to optimize the forward pass but show how to compute full SGD in ZK-SNARKs and optimize the softmax.

## 8 CONCLUSION

In this work, we have shown the feasibility of trustless audits for image classification and simple recommender systems. These audits include censorship to copyright audits. Although promising, much work remains to scale ZKAUDIT to larger models and datasets. We hope that ZKAUDIT can serve as inspiration for further research in audits, given the rise of API-gated models.

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

## A ZK-SNARKS

In this section, we describe how general-purpose ZK-SNARK proving systems work.

### A.1 INTUITION

The general intuition behind a ZK-SNARK proving system is that *any* function can be computed by a polynomial of sufficient size, i.e., that polynomials are universal. Then, to prove a function, the ZK-SNARK proving system encodes the function as a polynomial, commits to the polynomial, and proves that the polynomial evaluation is "as expected."

Thus, the primary questions a ZK-SNARK proving system tackles is how the function is encoded as a polynomial (since different proving restrictions have constraints on what polynomials can be expressed), how to commit to the polynomial, and how to prove to the verifier that the polynomial is "as expected."

### A.2 EXPRESSING FUNCTIONS

Although ZK-SNARK proving systems fundamentally deal with polynomials (albeit with specific constraints), functions are expressed via "front ends." These front ends include R1CS and AIRs. In this work, we focus on AIRs.

As mentioned in Section 2, AIRs have a grid of values with three types of constraints: 1) equality constraints, 2) table constraints, and 3) polynomial constraints. We provide examples of how these can be used to perform ML operations. Throughout, we denote the number of columns in the grid as $N$.

**Sum.** The first example we consider is the sum of a fixed size vector $\mathrm{Sum}(\vec{x}) = \sum_i^n x_i$, where $n = N - 1$. We can lay out the elements of the vector and the result $z = \mathrm{Sum}(\vec{x})$ in a row as follows:

$$x_1 | \cdots | x_n | z.$$

The constraint is:

$$z - \sum_i^n x_i = 0.$$

**Dot product without bias.** Consider a dot product of fixed size without a bias. Namely,

$$\mathrm{DotProd}(\vec{x}, \vec{y}) = \sum_i^n x_i \cdot y_i.$$

For the gadget, we let $n = \lfloor \frac{N-1}{2} \rfloor$.

To compute the dot product, we lay out $\vec{x}$ and $\vec{y}$ and the result $z = \mathrm{DotProd}(\vec{x}, \vec{y})$ as follows:

$$x_1 | \cdots | x_n | y_1 | \cdots | y_n | z$$

If $N$ is even, we leave a cell empty. The constraint is simply:

$$z - \sum_i^n x_i \cdot y_i = 0.$$

Suppose we had two vectors $\vec{x}$ and $\vec{y}$ of cardinality $m > n$. We can decompose the overall dot product into $\lceil \frac{m}{n} \rceil$ dot products. We can then use the sum gadget from above to add the partial results. As we will see, there are many ways to perform a large dot product.

**Dot product with bias.** Consider a dot product of fixed size with a bias: $\mathrm{DotProd}(\vec{x}, \vec{y}, b) = b + \sum_i^n x_i \cdot y_i$. Here, $n = \lfloor \frac{N-2}{2} \rfloor$. We can lay out the row as follows:

$$x_1 | \cdots | x_n | y_1 | \cdots | y_n | b | z$$

and use the constraint

$$z - b - \sum_i^n x_i \cdot y_i = 0.$$

In order to compose a larger dot product, we can decompose the dot product into $\lceil \frac{m}{n} \rceil$ dot products with biases. The first bias is set to zero and the remainder of the biases are set to the accumulation. This method of computing a larger dot product does not require the sum gadget.

As shown in this example, there are a number of ways to perform the same operation. The efficiency will depend on a large number of factors, including the total size of the circuit and the size of the dot products.

**ReLU.** As a final example, consider computing the ReLU function pointwise over a vector $\vec{x}$. Here, $|\vec{x}| = \lfloor \frac{N}{2} \rfloor$. We can simply lay out the row as

$$x_1 | \text{ReLU}(x_1) | \cdots | x_n | \text{ReLU}(x_n)$$

The constraints ensure that pairs of columns $(x_i, \text{ReLU}(x_i)) \in T$ for a table $T$ that contains the domain and range of the ReLU function. Other pointwise non-linearities can be performed similarly.

### A.3   COMMITTING TO POLYNOMIALS

Once the function is expressed in the frontend, the ZK-SNARK proving system must turn the frontend into a series of polynomial commitments. At a high level, these polynomial commitments (and their openings) consist of the proof. A full discussion of turning frontend expressions to polynomial commitments is outside of the scope of this work and we defer to Thaler et al. (2022) for an extended discussion. We discuss the high level ideas here.

There are many forms of polynomial commitments. The two we focus on in this work are the KZG Kate et al. (2010) and IPA (inner-product argument) Bünz et al. (2021) commitments.

The KZG commitment scheme requires access to a *structured reference string* (SRS), which must be generated securely (i.e., with no single party knowing how the string was generated). This structured reference string is essentially the powers of a random element in the field, where the random element is not known to anyone. Fortunately, an SRS has already been generated for the KZG commitments with 75 active participants PSE (2023).

The IPA commitment scheme is transparent, meaning that there need not be a trusted setup. However, it results in larger proofs. As a result, we focus on the KZG commitment scheme in this work, but note that IPA works as well depending on the application at hand.

## B   PROTOCOL

### B.1   REMOVING VERIFIED RANDOMNESS

In order to remove verified randomness, we further require the assumption that the hash function acts as a random oracle Bellare & Rogaway (1993). Under this assumption, the prover first computes a Merkle tree Merkle (1988) of the committed data using the random oracle hash function. The root of the Merkle tree then acts as random bits. To compute more random bits, the prover can repeatedly hash the Merkle root.

### B.2   PREVENTING TRAVERSAL ORDERING ATTACKS

Although ZKAUDIT does not prevent data poisoning attacks, ZKAUDIT can prevent traversal ordering attacks Goldwasser et al. (2022), in which a model undetectably changes its behavior on specific inputs. In order to do so, we force the model trainer to commit to using verified randomness in the future. Under the random oracle model (see above), we can also use the Merkle root to set the data traversal ordering.

### B.3   FORMAL SECURITY ANALYSIS

We provide a formal security analysis of ZKAUDIT, showing that ZKAUDIT is knowledge sound and zero-knowledge. ZKAUDIT requires two cryptographic primitives: a ZK-SNARK with standard properties (computational knowledge soundness, zero knowledge) and a random oracle hash function.

We refer to Bitansky et al. (2017) for an extended discussion of ZK-SNARKs and Menezes et al. (2018) for an extended discussion of hash functions.

Denote the verifier to be $V$ and the set of accepting inputs and witnesses to be $(x,w) \in \mathcal{R}$. The property of computational knowledge soundness guarantees that for every non-uniform probabilistic polynomial time (NUPTT) adversary $\mathcal{A}$, there exists an NUPTT extractor $\text{ext}_{\mathcal{A}}$ such that for all $\lambda$

$$P[(x,\pi) \leftarrow \mathcal{A}, w \leftarrow \text{ext}_{\mathcal{A}} : (x,w) \notin \mathcal{R} \wedge V(x,\pi) = 1] = \text{negl}(\lambda)$$

where we omit the dependence on the shared setup for brevity Atapoor & Baghery (2019). We use a strong version of zero knowledge called perfect special honest-verifier zero knowledge (PSHKZK) zcash (2022). Namely, for two NUPTT adversaries $\mathcal{A}_1, \mathcal{A}_2$, there exists a polynomial time simulator $\mathcal{S}$ such that

$$P[\mathcal{A}_1(x,\text{tr}) = 1 | (x,w) \leftarrow \mathcal{A}_2, \text{tr} \leftarrow \langle \mathcal{P}(x,w), V(x,\rho) \rangle] = P[\mathcal{A}_2(x,\text{tr}) = 1 | (x,w) \leftarrow \mathcal{A}_2, \text{tr} \leftarrow \mathcal{S}(x,\rho)]$$

where tr is the transcript and $\rho$ is the internal randomness of the verifier.

**Knowledge soundness of ZKAUDIT.** We first consider if it is possible for the model provider to produce a transcript of ZKAUDIT that is invalid. Under the random oracle hash function assumption, all outputs from the hash function are random. As such, the model trainer cannot modify the lexico-graphical ordering of the hashes. Then, the model trainer produces $T$ ZK-SNARKs, one for each step of the SGD. The soundness error is amplified by $T$ by the union bound. However, $T \cdot \text{negl}(\lambda)$ is still $\text{negl}(\lambda)$. Thus, the entire protocol is knowledge-sound.

**Zero-knowledge of ZKAUDIT.** We then consider if it possible for the verifier to extract information from the proofs and hashes. Since we use the random oracle assumption, it is computationally infea-sible for an attacker to learn anything about the pre-images of the hash of the model's weights, which is the only public information in ZKAUDIT-T. To avoid dictionary attacks against the hash, we use a random salt (i.e., blinding factor) to the model weights.

Furthermore, since we do not modify the underlying protocol of halo2, we inherit its zero-knowledge property. Namely, a computationally bounded adversary can learn nothing about the private inputs from the publicly revealed data.

**Knowledge soundness to bits of security.** When deployed in practice, we must use concrete instan-tiations of ZK-SNARKs and hash function. In particular, hash functions do not satisfy the random oracle model in practice. When performing an analysis of the "bits of security" of ZK-SNARKs and hash functions, we are interested in the computational soundness of the protocols, since the hashes and proofs are published.

Let the hash function have $\lambda_1$ bits of security and the ZK-SNARK protocol have $\lambda_2$ bits of security. Let $\lambda = \min(\lambda_1, \lambda_2)$. We are interested in the worst case analysis where a single compromised hash or ZK-SNARK proof compromises the entire protocol. By the union bound, the bits of security after $D + 4T$ hashes and proofs is $\lambda/(D+4T)$.

In practice, $D < T$, so we can upper bound the amplification by $5T$. Furthermore, $T$ is often in the order of millions, which decreases $\lambda$ by around 25 bits. As such, to achieve 100 bits of security of ZKAUDIT, we require $\lambda > 125$. In practice, halo2 has 128 bits of security and the Poseidon hash function (which we can use) has parameters for 128 and 256 bits of security.

## C   FURTHER OPTIMIZATIONS

In addition to the optimizations described in the main text, we describe several additional optimiza-tions to improve performance.

**Packing.** The maximum size of a lookup table in a halo2 circuit is bound by the number of rows is the circuit. As a result, if the size of the lookup table is larger than the size necessary for a single backwards pass, we can pack multiple backward passes in a single circuit. This is particularly useful for circuits with high scale factors. We implemented this optimization for all circuits used in the evaluation.

**Folding.** Since SGD repeats the same computational pattern many times, we can use *folding* schemes for proving ZK-SNARKs Kothapalli et al. (2022). Folding schemes are especially useful for repeated

computations. Although we have not implemented this optimization, we expect the costs to be reduced by $10\times$ or more.

## D   ADDITIONAL EVALUATION INFORMATION

### D.1   EVALUATION SETUP

**Recommender system model.**   We use the following model for the recommender system:

1. An embedding dimension of 100 for both movies and users.
2. The embeddings are then concatenated.
3. The concatenated embeddings are fed through two fully connected layers. The first fully connected layer has an output dimension of 128 and the second has an output dimension of 1. The first fully connected layer has a relu6 nonlinearity.

We used the mean-squared error loss.

**Hardware.**   We use the Amazon Web Services (AWS) `g4dn.8xlarge` instance type for all experiments. We use the cost of a spot instance ($1.286 per hour at the time of writing) to estimate costs.

**Code.**   We have anonymized our code here: `https://anonymous.4open.science/r/zkml-72D8/README.md`

## E   FURTHER EVALUATIONS

We further evaluated ZKAUDIT-T on CIFAR-10 and MNIST. As a comparison, we took the models from Rathee et al. (2023). Rathee et al. (2023) is in the *semi-honest* multi-party computation setting, which does not handle malicious adversaries. Furthermore, it does not produce audits.

We compared the cost producing the proofs for ZKAUDIT-T to *just* the bandwidth costs for Rathee et al. (2023). For CIFAR-10 HiNet, ZKAUDIT-T costs $2,417 compared to $475,234 for Rathee et al. (2023). As we can see, ZKAUDIT-T is $196\times$ cheaper than Rathee et al. (2023).

## F   END-TO-END TWITTER EXAMPLE

As mentioned, complete audits require combining ZKAUDIT with verified data access and other forms of verified computation. For example, in the Twitter audit example, the auditor must also be convinced the input data is valid and that the preprocessing of the data to the ML model input format is done correctly. In order to ensure the end-to-end validity, we can combine ZKAUDIT with other methods of verified computation and data access:

1. To ensure validity of data access, we can use techniques such as vSQL Zhang et al. (2017).
2. To ensure validity of preprocessing, we can use techniques such as RISC-Zero or ZK-SNARKs for general purpose computation Arun et al. (2023).

