# OpenReview forum: "Trustless Audits without Revealing Data or Models"
_ICLR.cc/2024/Conference — Submitted to ICLR 2024_

### Official Review · Reviewer_Ww1n · 2023-10-29

**Soundness:** 2 fair
**Presentation:** 2 fair
**Contribution:** 3 good
**Rating:** 6
**Confidence:** 4

**Summary:**

This paper proposes a method for privately auditing machine learning models as well as their training data based on the use of cryptographic commitments and zero-knowledge proofs. More precisely, the audit framework proposed is composed of two phases: one in which the training data as well as the model weights are cryptographically committed and one in which an audit function F is computed on the data along with a zero-knowledge proof of the result. To realize this, a novel zero-knowledge protocol for computing the backward pass of the training of a neural network is also proposed.

**Strengths:**

The paper proposes an interesting approach to be able to audit machine learning models such as neural networks in a privacy-preserving manner. In addition, the use of ZK-SNARKS provides strong security and privacy properties.

One of the main novelty of the paper is the design of an approach for computing the backward pass of a stochastic gradient descent algorithm based on ZK-SNARK. Several optimisation tricks are also proposed to be able to make the approach more efficient and control the computation-utility trade-off. Overall, this enables to prove the training in privacy-preserving manner while previous works were only focusing on inference.

Overall, the paper is well-written although the addition of an outline at the end of the introduction would help to clarify its structure.

**Weaknesses:**

In the introduction, there is a bit of confusion between the issue of training the model privately using secure multiparty computation vs performing an audit in a collaborative manner. The description of ZK-SNARKs in Section 2 is also a bit too concise for a reader that is not already familiar with this concept. I suggest to add a few concrete examples of what x and w could be, in particular within the context of privacy-preserving machine learning. Similarly, an intuitive example would help to understand what information or properties could be encoded in the 2D grid of arithmetic intermediate representations. Similarly more details are needed to understand the concepts of KZG commitments, inner-products arguments or structured-reference strings. Otherwise, the paper is going to be difficult to follow for a reader that does not have already a strong cryptographic background

The limit on which functions can (or cannot) be audited with the proposed approach is not clear. For instance, it is not clear for me if the proposed approach could be used to prove fairness properties about the model. A generic discussion on the applicability but also limits of the method would help.

The experiments are conducted on a limited number of datasets and architectures and thus it is not clear how the approach would scale. For instance, I suggest to the authors to provide additional experiments with classical datasets such as MNIST and CIFAR and architecture such as Resnet. In addition, one shortcoming of the current experiments is that they only report the training accuracy but not the test accuracy. The cost should be also reported in terms of computational time in addition of the monetary cost to be more meaningful.

**Questions:**

-The notion of trust should be more specifically defined in the paper, in particular as the focus of the paper is to argue that the proposed approach is « trusttless ».
-How does a traversal ordering attacks works?
-What are the test accuracies obtained for the different conducted experiments?
-How does the use of salt to prevent dictionary attacks when hashing the weights combine with the ZK-SNARKS?

---

> ### Author Response · Authors · 2023-11-15
>
> Dear reviewer,
>
> Thank you for your detailed comments. We have conducted additional experiments on CIFAR-10 and MNIST and have updated the draft to address your concerns.
>
> ## Presentation of ZK-SNARKs
>
> We have incorporated your suggestions on the presentation of ZK-SNARKs into the updated draft. Due to the limited space, we present an extended discussion in the Appendix, including four examples of using AIRs and further discussion on ZK-SNARKs.
>
>
> ## Limit on audit functions
>
> In principle, any computable function can be used as the audit function. Fairness properties can easily be checked with the same overhead as the copyright audit. We have updated the draft to clarify this point (Section 6).
>
>
> ## Experiments on CIFAR-10, MNIST
>
> We compared our method to semi-honest MPC published this year [1]. Note that this semi-honest MPC is not secure against malicious adversaries. We computed the proving times and costs for the semi-honest MPC and ZkAudit. For the semi-honest MPC, we _only_ measured the egress fees.
>
> CIFAR-10 HiNet
>
> [1]: 100,827 minutes, 5,280,375 GB, 475,234 dollars
>
> ZkAudit: 113,311 minutes | 2,417 dollars
>
> MNIST FFFNN
>
> [1]: 626 minutes, 30,820 GB, 2773.8 dollars
>
> ZkAudit: 13,226 minutes, 282 dollars
>
>
> Our method is up to 196x cheaper than secure MPC and takes approximately the same amount of time to prove on CIFAR10. Although MNIST is slower, it is substantially cheaper.
>
> [1]  https://www.usenix.org/system/files/sec23fall-prepub-212-rathee.pdf
>
>
> ## Test accuracy
>
> All accuracies reported are the _test_ accuracy after the training procedure. We have clarified this in the draft (Section 5).
>
>
> ## Notion of trust
>
> In this work, trust means that a third party must assume the model provider acts according to the protocol. A malicious adversary may deviate from the protocol. Under mild cryptographic assumptions, ZkAudit does not require the model provider to act honestly as the model provider will be caught if attempting to deviate from the protocol.
>
>
> ## Traversal ordering attack
>
> A traversal ordering attack allows a trainer to change the model outputs in undetectable ways by simply changing the order the data is seen at training time [1].
>
> [1] https://arxiv.org/abs/2204.06974
>
>
> ## Salt and dictionary attacks
>
> Denote the weights of the model as W. A salt appends _random_ bits R to W. Then, under mild assumptions on hash functions, H(W || R) is not susceptible to dictionary attacks, since the padding R is random (so the output of the hash will appear random).

---

> > ### Comment · Reviewer_Ww1n · 2023-11-22
> >
> > I thank the authors for their responses that have clarified some of the points I had. Thus, I am willing to upgrade a bit my rating to the paper.

---

> > > ### Author Response · Authors · 2023-11-23
> > >
> > > Dear Reviewer Ww1n,
> > >
> > > Thank you for the clarification! Would you be willing to update your review with an increased score?
> > >
> > > Best,
> > > #2901 authors

---

### Official Review · Reviewer_TAmM · 2023-10-29

**Soundness:** 4 excellent
**Presentation:** 3 good
**Contribution:** 3 good
**Rating:** 8
**Confidence:** 3

**Summary:**

The paper introduces a new method, ZkAudit, for enabling auditing of deep learning models without revealing data or weights using zero-knowledge proofs. The method is separated into two parts: ZkAudit-T, which can be used to prove that a particular set of weights was the result of training with SGD on a particular dataset; and ZkAudit-I, which can be used to verifiably compute an audit function on the datasets and weights. The methods are all based on ZK-SNARKs. The authors implement several optimizations to make the method work with SGD: rounded division, an improved softmax function, and lowered precision throughout the network. In benchmarking on both an image classification task and a recommender system task, the authors show that ZkAudit can be implemented with relatively low cost compared to other methods for audit questions such as copyright censorship detection.

**Strengths:**

The paper’s primary strength is in its identification of an important problem - the reluctance of model owners to share their models for auditing purposes - and the optimizations that the authors implement to improve the method’s performance over existing work. The evaluation on two different kinds of tasks also shows the promise of the method overall.

**Weaknesses:**

The primary weakness of the paper is that, while it shows that the technique has some promise, it is difficult to tell how practical the implementation would be in a real-world setting. The paper shows improvements in cost and performance over existing methods, but there is no comparison to non ZK-based methods. The paper could be improved by showing how far away ZkAudit is from the cost/time it would take if the auditors had access to the model and dataset or even if they were in an SMPC setting as the authors cite early in the paper. While I expect that ZkAudit will be slower or more costly than these other scenarios, it would be useful to know by how much - is it several orders of magnitude? Additionally, I believe the paper needs more discussion of the limitations of the method and how that would implement the practicality of the method. In the next questions section, I have some questions that could help the authors expand on this discussion.

**Questions:**

In the results sections, the authors show the cost of various experiments they ran. Can the paper provide more reference points for how these compare to prior work and alternative methods?

The fact that the model architecture cannot be shared seems like a major limitation. How does this restrict the types of audit questions and functions that can be shared?

The method is set up to not reveal the training data and weights, but it does reveal the output of the audit function $F$. Are there ways to limit the amount of information that the audit function reveals? For example, could I pass an audit function that would leak information that the owner would not want to share? Or does the fact that $F$ needs to be implemented as a ZK-SNARK prevent that?

One very common audit task is to detect demographic disparities in model performance. I think answering this question would be precluded by the fact that model architecture is not shared, but I am curious to hear more discussion on this task and how it fits in with ZK-Audit.

How would an output privacy method like differential privacy stack up against ZK-Audit in terms of its capabilities and tradeoffs? It would be nice to see some discussion of what ZK-Audit offers over those methods (for example, I imagine that there is no way to verify that a particular model was trained on a particular dataset with differential privacy, even if you released noisy weights and data).

---

> ### Author Response · Authors · 2023-11-15
>
> Dear reviewer,
>
> Thank you for your detailed comments. We have responded to the questions below and have updated our paper to address your questions.
>
> ## Cost comparison to secure MPC
>
> _Secure_ MPC is unfortunately slow and expensive. For example, work published this year on semi-honest, secure MPC, which is not even secure against malicious adversaries, takes _100,827 minutes_ and _5,280,375 gigabytes_ of communication to train _on epoch_ on CIFAR10 [1]. The data egress fees alone would cost _$475,233.75_.
>
> We took the models in [1] and compared the total cost of _just_ data egress fees compared the cost of computing the ZK-SNARK proofs for CIFAR10.
>
> CIFAR-10 HiNet
>
> [1]: 100,827 minutes, 5,280,375 GB, 475,234 dollars
>
> ZkAudit: 113,311 minutes | 2,417 dollars
>
> MNIST FFFNN
>
> [1]: 626 minutes, 30,820 GB, 2773.8 dollars
>
> ZkAudit: 13,226 minutes, 282 dollars
>
> Our method is up to 196x cheaper than secure MPC and takes approximately the same amount of time to prove on CIFAR10. Although MNIST is slower, it is substantially cheaper.
>
> [1] https://www.usenix.org/system/files/sec23fall-prepub-212-rathee.pdf
>
>
> ## Cost comparison to prior work
>
> _No_ prior work for ZK-SNARKs for ML can perform training, since no prior work can compute the softmax function. We are the first method to be able to perform training on modern DNNs.
>
> To compare the cost of prior work, we removed the softmax and estimated the time for the remainder of the operations. Even without the softmax, the prior work is up to 94x costlier than our proposed method.
>
>
> ## Model architecture
>
> The model architecture must be shared with the verifier. We have clarified this point in the updated draft (Section 3).
>
>
> ## Audit function
>
> The audit function itself must be computed within a ZK-SNARK for full security (this is the purpose of ZkAudit-I). The model provider can choose to participate in an audit. So, if an audit leaks sensitive information (such as the model weights itself), the model provider can choose to not participate in the audit. Moreover, note that the ZK-SNARK can be configured to reveal only whether an audit passes or not, so that only a single bit of information is revealed to the verifier. We have clarified this point in the updated draft (Section 6).
>
>
> ## Demographic disparities
>
> Computing demographic disparities _only_ requires access to the outputs of the model, not the model architecture itself. We can compute demographic disparities with the same cost as the copyright audit.
>
> Nonetheless, ZkAudit requires the architecture to be shared. We have clarified both points in an updated draft (Section 6).
>
>
> ## Comparison to differential privacy
>
> As you point out, there is no way to verify that a model was trained on a particular dataset with differential privacy. Differential privacy is thus orthogonal to our work and can be even combined with ZkAudit-T.

---

> > ### Comment · Reviewer_TAmM · 2023-12-04
> > **Thank you**
> >
> > Given the authors' responses and clarifications on the limitations of the work, I am upgrading my score slightly from 6 to 7. I do think this work should be published, as it seems the limitations are not as severe as I thought they were.

---

### Official Review · Reviewer_uRaA · 2023-11-04

**Soundness:** 3 good
**Presentation:** 2 fair
**Contribution:** 2 fair
**Rating:** 6
**Confidence:** 2

**Summary:**

The work describes methods with which a model can be trained in a manner that can later be
scrutinized in a zero-knowledge manner. That is, the training (and inferences) results in a proof
with which the trainer can prove to verifiers (presumably users who want to perform inferences but
cannot have the model itself) any desired property about the data, training, (and inference) they
might be interested in like presence of undesired data in the training dataset, without revealing
anything else about the training data or model parameters. Importantly, there are no trust
assumptions with regards to the prover. To achieve this (with reasonable performance), the work
introduces number representations with configurable precision/scale to be used in the ZK-SNARK
approach which are required for the backwards network passes in the training process. The paper
also describes a performant means of computing softmax.

Experimental evaluations using relatively small recommender and image classification models are
then presented showing: (1) scalability of the approach in terms of the numerical representation
precision, (2) accuracy/cost tradeoffs where the cost is a function of numerical precision, and (3)
relative loss of precision as compared to typical 32-bit floating point models (i.e. baselines).
Three types of queries or "audits" are also demonstrated alongside their estimated costs which are
said to be within "reason" though I cannot tell what sorts of costs are to be expected.

**Strengths:**

+ Work makes it possible to audit both training and inference as well as any audit-relevant
  property of data and the trained model without revealing either.

+ Quite general auditing capabilities. All three auditing examples are compelling.

**Weaknesses:**

- Novelty/contribution may be limited (or unclearly stated). Backwards pass work includes numerical
  representation adjustments and a better implementation of softmax. The structure of the paper
  thus makes it looks like these two bits of work were the missing pieces of zk-SNARK work that
  prevented them from being realistically applied to SGD. These are presumed to be "optimizations"
  as in the related work statement:

    "All of the work we are aware of focuses on optimizing inference, so all of this work omits the
     softmax computation and does not optimize the backward pass. In this work, we leverage ideas
     from inference to optimize the forward pass but show how to compute full SGD in ZK-SNARKs and
     optimize the softmax."

  This suggests that while other works can perform SGD, its backwards pass, and/or softmax, but
  just not efficiently. The title of the work and other statements, however, suggests that this is
  the first work that can handle these computations at all.

  Further, the contributions in the numerical representations do not seem significant and softmax
  may be even omitted from model training (or inference) without significant changes to training or
  inference. There thus appears some incongruity with the stated contribution and methods achieving
  those contributions.

  Suggestion: clarify whether the work presents the first verification systems that includes
  data/model training and that to achieve this within the frameworks from prior works, only changes
  to numerical representation (and softmax) were needed. Alternatively, if there was more necessary
  to achieve the goal, describe them in more detail. Alternatively, it may be that prior works
  already achieved the goal with less performant system in which case the title of the work and
  contributions need to be clarified. In this last case, comparisons of accuracy and efficiency
  need to be provided (as in Figures 1, 2).

Smaller thing / suggestions.

- Model parameters are not revealed to verifiers but model architectures are. This is noted but the
  abstract is unqualified, please adjust the abstract to make this clear.

- Some tabular results do not state which operation they are measuring and whether it is of a
  single instance of that operation (for inference) or multiple / entire dataset. Please include
  this.

- For prior techniques which do not handle softmax at all or well, consider including versions of
  models without softmax alongside the softmax ones. By this I don't mean to merely use an existing
  softmax implementation as done in Section 5.3, but instead remove the entire softmax operation
  from the model.

- Table 3: include units in table or description.

- Some results that depend on dataset size could be better presented in terms of cost per instance
  (Copyright audit is one example). The currently noted "$108" is not

- Please include a bit more how operations are implemented via AIR in its background section.

**Questions:**

- Question A: With regards to the weakness noted in the weaknesses section, please provide noted
  clarifications.

- Question B: How does this work compare: Efficient Representation of Numerical Optimization
  Problems for {SNARKs}. Angel et al. 2022. I also see several papers in federated learning that
  leverage SNARKs for enhancing trust. If models involving softmax or similar steps to the
  presently-experimented ones were used, I imagine they would have similar accuracy problems. Are
  their solutions applicable?

- Question C: Can the numerical representation work described in the paper by applied to other
  verified computation problems beyond model training?

---

> ### Author Response · Authors · 2023-11-15
>
> Dear reviewer,
>
> Thank you for your detailed comments. We have addressed them in an updated version of the paper and summarized our changes along with answers to your questions below.
>
> ## Requiring the softmax
>
> All methods we are aware of for doing multi-class classification require performing a complex non-linearity such as a softmax or multi-class hinge. Thus, _no_ prior work is able to perform training. For example, it is unclear how to perform even the exponential function in the prior work for ZK-SNARKs for ML.
>
> Inference can be done by removing the softmax layer and taking the argmax of the logits instead, which is what prior work does. ZkAudit-I leverages prior work. We have made these points more clear in the paper.
>
>
> ## Comparison to Angel et al (2022)
>
> The “numerical optimization problems” Angel et al (2022) address are constrained optimizations via linear programming and semidefinite programming. Neither of these is applicable to DNN training, which we focus on. In fact, Angel et al do not show how to perform the forward pass or backward pass for DNNs. We have included a discussion in the related work section.
>
>
> ## Does our work apply to other verified computation problems?
>
> Yes, our work applies to other verified computation problems. Our work is most applicable to optimization problems that need to perform complex non-linear functions as part of their optimization (e.g., the softmax, exponentials).
>
>
> ## Assorted suggestions
>
> We have addressed your smaller suggestions:
> 1. We have updated the abstract to mention model architectures are revealed.
> 2. We updated table captions to mention if the results were for a single operation or for the entire dataset.
> 3. We do compare our work against prior work estimates when removing the softmax. However, training would not work without the softmax.
> 4. We have included units in Table 3.
> 5. We have clarified how the audits scale with the size of the dataset.
> 6. We have included more information about how operations are implemented within the AIR in the background section.

---

> ### Comment · Reviewer_uRaA · 2023-12-05
>
> Thank you for the clarifications. I will adjust my score to marginally above threshold. My remaining concern is about novelty/level of contributions which is a bit subjective and will have to be resolved by the paper/area chair.

---

### Meta-Review · Area_Chair_WRAN · 2023-12-05

**Metareview:**

The paper studies how to trustlessly audit a machine learning model’s and training data’s properties while keeping the model weights and training data secret. The paper proposes a protocol, ZKAudit, whereby model providers publish cryptographic commitments of model weights and datasets, along with a zero-knowledge proof certifying that the published commitments are indeed derived from training the model. Then, on audit requests, the model providers can reply by computing a function of the training dataset or model weights and then releasing the output of this function along with a zero-knowledge proof certifying the correct execution of this function. The main contributions of the paper are new methods for computing zero-knowledge proofs for stochastic gradient descent on neural networks for simple recommender systems and image classification models. Specifically, the paper builds on the recent work of Kang et al., who have recently optimized the forward pass for int8 inference in zero-knowledge succinct non-interactive arguments of knowledge (ZK-SNARKs). The current paper shows how to compute the backward pass of stochastic gradient descent in a ZK-SNARK. Moreover, the paper’s method operates in fixed-point arithmetic and includes an optimization for a high-performance softmax in ZK-SNARKs. The paper also introduces two techniques for computing ZK-SNARKs for gradient descent, and which improve the accuracy of training: rounded division in finite field ZK-SNARK constraints and variable precision fixed-point arithmetic.

The question tackled by this paper is interesting and well-motivated. The main downside of the paper is that its novelty is limited for ICLR. In particular, the main improvements of the paper, while definitely worthy of being published in some venue, can be seen as implementation tricks that likely won’t be of interest to a broad audience at ICLR. Therefore, I feel that the paper might be a better fit for another conference (e.g., one on applied cryptography).

**Justification For Why Not Higher Score:**

The main improvements of the paper, while definitely worthy of being published in some venue, can be seen as implementation tricks that likely won’t be of interest to a broad audience at ICLR.

**Justification For Why Not Lower Score:**

n/a

---

### Decision · Program_Chairs · 2024-01-16

Reject